# mRNA COVID-19 vaccine booster fosters B- and T-cell responses in immunocompromised patients

Elena Azzolini[1,2,*], Chiara Pozzi[2,*], Luca Germagnoli[2], Bianca Oresta[3], Nicola Carriglio[3], Mariella Calleri[3], Carlo Selmi[1,2], Maria De Santis[1,2], Silvia Finazzi[2], Carmelo Carlo-Stella[1,2], Alexia Bertuzzi[2], Francesca Motta[1,2], Angela Ceribelli[1,2], Alberto Mantovani[1,2,4], Fabrizio Bonelli[3], Maria Rescigno[1,2]

**SARS-CoV-2 vaccination has proven effective in inducing an immune response in healthy individuals and is progressively us allowing to overcome the pandemic. Recent evidence has shown that response to vaccination in some vulnerable patients may be diminished, and it has been proposed a booster dose. We tested the kinetic of development of serum antibodies to the SARS-CoV-2 Spike protein, their neutralizing capacity, the CD4 and CD8 IFN-γ T-cell response in 328 subjects, including 131 immunocompromised individuals (cancer, rheumatologic, and hemodialysis patients), 160 health-care workers (HCW) and 37 subjects older than 75 yr, after vaccination with two or three doses of mRNA vaccines. We stratified the patients according to the type of treatment. We found that immunocompromised patients, depending on the type of treatment, poorly respond to SARS-CoV-2 mRNA vaccines. However, an additional booster dose of vaccine induced a good immune response in almost all of the patients except those receiving anti-CD20 antibody. Similarly to HCW, previously infected and vaccinated immunocompromised individuals demonstrate a stronger SARS-CoV-2–specific immune response than those who are vaccinated without prior infection.**

## Introduction

From December 2020 several anti-SARS-CoV-2 vaccines have been approved by the drug authority agencies for emergency use for the prevention and management of COVID-19. SARS-CoV-2 vaccination has proven to be effective in protecting against hospitalization and death in Israel (Haas et al, 2021), and, as shown by the COVID-19 vaccine breakthrough infection surveillance, also in the United States even towards the Delta variant (Scobie et al, 2021). This indicates that vaccines can help control COVID-19 severity and the pandemic itself. Indeed, all of the vaccines approved so far have proven great efficacy in activating an immune response in healthy individuals (Polack et al, 2020; Walsh et al, 2020; Abu Jabal et al, 2021; Dagan et al, 2021; Haas et al, 2021; Voysey et al, 2021; Arunachalam et al, 2021a, 2021b), and we and others have shown that one dose is sufficient in boosting the immune response in SARS-CoV-2 previously exposed subjects (Levi et al, 2021; Saadat et al, 2021; Sadoff et al, 2021; Samanovic et al, 2021; Krammer et al, 2021a, 2021b Preprint). However, the ability of mRNA-based SARS-CoV-2 vaccines to immunize primary or treatment-induced immunocompromised individuals has recently been questioned (Collier et al, 2021). In particular, patients with inflammatory bowel disease under infliximab treatment (Kennedy et al, 2021), patients who have received an allogeneic stem cell transplantation (Lafarge et al, 2022), cancer patients (Chung et al, 2021; Ribas et al, 2021; Zeng et al, 2021; Greenberger et al, 2021a, 2021b; Thakkar et al, 2021a, 2021b), methotrexate treatment (Mahil et al, 2021), kidney transplant or hemodialysis (Bachelet et al, 2021; Danthu et al, 2021), or multiple sclerosis (Apostolidis et al, 2021) have all demonstrated a reduced ability to mount an immune response, potentially adversely affecting protection offered by vaccines. However, studies in which a comprehensive comparative analysis of both humoral and cellular immune responses after a third dose of vaccine is lacking.

Indeed, the type of immunomodulatory treatment may have a differential effect according to the immune cell which is targeted. For instance, B-cell–directed therapies for hematological malignancies have been shown to affect the production of antibodies in response to SARS-CoV-2 vaccination because of B-cell depletion and/or disruption of the B-cell receptor signaling pathway while leaving unaltered the T-cell response (Apostolidis et al, 2021). This T-cell response may compensate for the B-cell response and may explain why anti-CD20–treated patients are still protected from COVID-19 (Bange et al, 2021). By contrast, a general immune suppression due to drug treatments or the disease itself may affect both humoral and cellular responses. Hence, it is very important to evaluate the immunization status and the duration of response in immunocompromised patients undergoing SARS-CoV-2 vaccination

[1]Department of Biomedical Sciences, Humanitas University, Pieve Emanuele MI, Italy   [2]Istituto di Ricovero e Cura a Carattere Scientifico (IRCCS) Humanitas Research Hospital, Rozzano MI, Italy   [3]DiaSorin S.p.A., Saluggia VC, Italy   [4]William Harvey Research Institute, Queen Mary University of London, London, UK

Correspondence: maria.rescigno@hunimed.eu
*Elena Azzolini and Chiara Pozzi contributed equally to this work.

and relate it to the type of treatment. Here, we compared the antibody production, CD4 and CD8 T-cell response to the vaccine Spike protein, as well as the neutralization potential of the antibody response in response to two or three doses of SARS-CoV-2 vaccine in 328 subjects including health-care workers (HCW), elderly subjects (>75 yr), and immunocompromised patients with different pathologies either in hemodialysis, with cancer or rheumatological diseases in relation to their treatments.

We show that one of the major determinants of a successful immune response was the immune status, exposure to SARS-CoV-2 infection and type of treatment at the time of vaccination and that three doses of vaccine allowed achieve immunization even in immunocompromised individuals. However, as expected, anti-CD20 treatment impaired the development of an antibody response even after the third dose, suggesting that patients under this treatment should wait to receive the shots after interrupting the therapy. Patients under mycophenolate also respond poorly to vaccination, but interruption of therapy for just 1 wk allows activation of the immune response. We also show that SARS-CoV-2–recovered immunocompromised individuals, similarly to healthy subjects (Levi et al, 2021; Saadat et al, 2021; Sadoff et al, 2021; Samanovic et al, 2021; Krammer et al, 2021a, 2021b Preprint), achieved a strong immune response, quicker than naïve subjects. Overall, this study highlights a need in a booster dose of vaccine in immunocompromised individuals, which should however consider their immune status and treatment. SARS-CoV-2–recovered patients, instead, should be considered for the booster dose on an individual basis.

# Results

### Clinical study

In this observational study, we analyzed the antibody production, the CD4 and CD8 T-cell and the neutralizing antibody response to SARS-CoV-2 Spike protein in 328 subjects (Table 1), including health-care workers (n = 160), elderly people >65 yr (n = 37), and 131 immunocompromised patients with different pathologies including patients in hemodialysis (n = 53), with cancer (n = 30) or rheumatological disease (n = 48) at 2–4 mo (T3) after the second dose of mRNA SARS-CoV-2 vaccination (Spikevax or Comirnaty). For immunocompromised patients we investigated the humoral and cellular immune response also at 2 wk after the third (booster) dose (T4). In particular, 13 (44%) cancer patients, 31 (65%) patients with rheumatic disease and 44 (83%) patients in hemodialysis received the third dose. Moreover, for HCW and cancer patients we tested the kinetics of B- and T-cell development before vaccination (T0) at 21–28 d after the first dose (T1), 10–26 d after dose 2 (T2), and 2–4 mo (T3) after the second dose (Fig 1). 62 individuals had been previously exposed to SARS-CoV-2 (Table 1), and among these, only 6 of 18 (33%) cancer patients, 1 of 5 (20%) hemodialysis patients, and 1 (100%) rheumatic disease patient received the third dose. The immune response was correlated with the type of pathology, the immune status, and the treatment (Table 2).

### SARS-CoV-2–naïve cancer patients treated with anti-CD20 fail to produce neutralizing antibodies

SARS-CoV-2 particle internalization is mediated by the binding of the trimeric form of the Spike protein with the ACE-2 receptor on host cells (Hoffmann et al, 2020). We chose to test the level of IgG antibodies directed to the trimeric form of Spike protein (LIAISON SARS-CoV-2 TrimericS IgG; DiaSorin) to have a better correlation with neutralizing antibodies. Nevertheless, we also tested the neutralization ability of the ensued antibodies via a surrogate test of Spike neutralization (cPass; GenScript). As shown in Fig S1A, although the antibody response was induced in health-care workers already after the first vaccine dose (T1) and reached a climax 10 d after the second dose (T2), it was either undetectable in cancer patients receiving anti-CD20 treatment (blue triangles, category 2) or reduced in patients receiving other drugs with low/medium impact to the immune system (orange and green/yellow

**Table 1. Cohort design and summary statistics.**

| | HCW | Elderly ≥75 | Cancer patients | Rheumatic disease patients | Dialysis patients |
|---|---|---|---|---|---|
| **Subjects (n)** | **160** | **37** | **30** | **48** | **53** |
| Sex | | | | | |
| Female | 108 (67.5%) | 20 (54.1%) | 14 (46.7%) | 29 (60.4%) | 19 (35.8%) |
| Male | 52 (32.5%) | 17 (45.9%) | 16 (53.3%) | 19 (39.6%) | 34 (64.2%) |
| Age | | | | | |
| Mean (Min–Max) | 30.23 (19–77) | 79.03 (75–87) | 54.9 (35–79) | 54.92 (25–78) | 73.28 (50–93) |
| SARS-CoV-2 naturally infected | | | | | |
| No | 124 (77.5%) | 35 (94.6%) | 12 (40%) | 47 (97.9%) | 48 (90.6%) |
| Yes | 36 (22.5) | 2 (5.4%) | 18 (60%) | 1 (2.1%) | 5 (9.4%) |
| Vaccine Type | | | | | |
| Comirnaty Pfizer | 160 (100%) | 37 (100%) | 30 (100%) | 0 (0%) | 53 (100%) |
| Spikevax Moderna | 0 (0%) | 0 (0%) | 0 (0%) | 48 (100%) | 0 (0%) |

Demographic and clinical information, including age, sex, SARS-CoV-2 infection, and vaccine type.

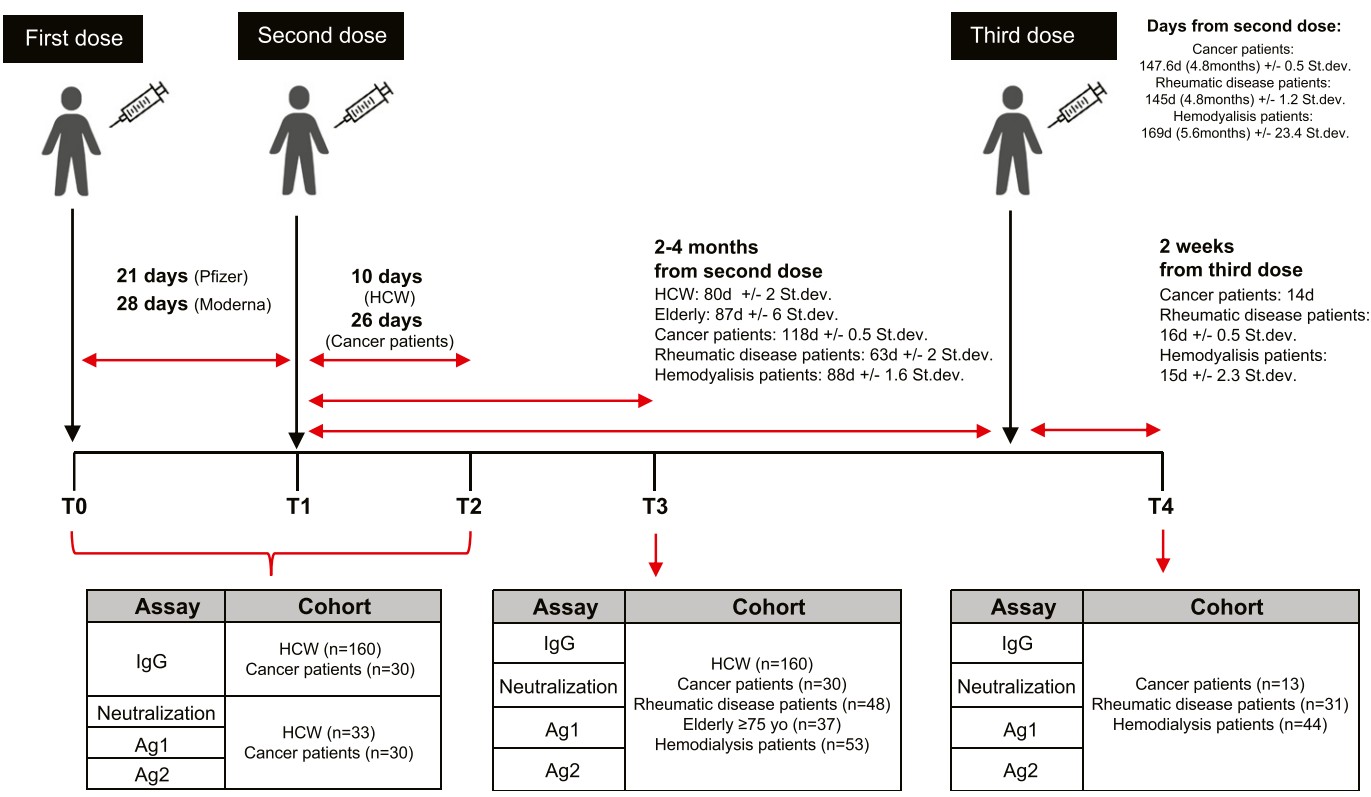

**Figure 1. Experimental design.**
IgG antibody response, the CD4 and CD8 T-cell activation (Ag1 and Ag2) and the neutralizing antibody response to SARS-CoV-2 spike protein developed after mRNA SARS-CoV-2 vaccination (Spikevax or mRNA-1273, Moderna—Comirnaty or BNT162b2, Pfizer-BioNTech) were analyzed as a part of two observational studies approved by the Ethical Committee of Istituto Clinico Humanitas, in compliance with the Declaration of Helsinki principles. The studies were conducted at Istituto Clinico Humanitas and comprises a longitudinal sample collection, including health-care workers (n = 160) and cancer patients (n = 30) and a cross-sectional sample collection, including elderly subjects (n = 37), patients with rheumatic diseases (n = 48), and patients in hemodialysis (n = 53). Immunocompromised patients received a third dose (booster) ±5 mo after the second dose. Analyzed time points were: the day of the first dose (T0), 21–28 d after the first dose (T1), 10–26 d after the second dose (T2), 2–4 mo after the second dose (T3), and 2 wk after the third dose (T4).

triangles, categories 0 or 1, respectively) at any time point between T0 and T3 (Fig S1A). In those patients that experienced an antibody response, the titers were much lower than those of the HCW suggesting that the amplitude of the antibody response was compromised. However, a booster dose of vaccine increased the antibody titers at levels similar to those of HCW, except for anti-CD20 treated cancer patients which remained undetectable (Fig S1A, T4). As the latter patients were discouraged to take a booster dose, we could test only three of eight patients who insisted to receive it. Wherever detectable, the antibodies were neutralizing and were preserved at least 4 mo after vaccination (T3), but only in those patients that were not in active treatment at the time of vaccination (orange triangles, Fig S1A and Table 2). By contrast, the antibodies raised in HCW were all neutralizing (Fig S1A). Regarding SARS-CoV-2 previously exposed individuals, whereas nearly all HCW required one single dose to reach a very strong neutralizing antibody response, as we and other previously described (; Levi et al, 2021; Saadat et al, 2021; Sadoff et al, 2021; Samanovic et al, 2021; Krammer et al, 2021a, 2021b Preprint), SARS-CoV-2 naturally infected cancer patients required two doses to reach comparable neutralizing antibodies (Fig S1B), but almost all of them (16 of 18) developed IgG antibody response, even if cancer patients were

under active treatment at the time of vaccination (12 of 18, Fig S1B and Table 2). In particular, 10 were treated with drugs belonging to category 0, one with drug of category 1 (green triangles, Doxorubicin+Cisplatin) and one with anti-CD20 (blue triangles, category 2) (Fig S1B and Table 2). A booster dose increased the amount of serum antibodies, particularly the neutralizing antibodies (Fig S1B, T4). The only naturally infected cancer patient under active anti-CD20 treatment did not increase antibodies even after the second dose (T2) (blue triangles, Fig S1B), and was advised to take a third dose after stopping the anti-CD20 treatment. A higher number of patients under this treatment is required to reach any conclusions.

### SARS-CoV-2–naïve cancer patients treated with anti-CD20 may fail to activate T-cell responses

The induction of a CD4 or CD8 T-cell response is an additional arm of an effective vaccination. We thus evaluated the kinetic of anti-Spike T-cell response activation in the two groups, by using specific CD4 (Ag1) and CD4 plus CD8 (Ag2) T-cell epitopes of the Spike protein. As shown in Fig S2A, we found that the T-cell response (both to Ag1 and Ag2) was low in general in cancer patients and was

**Table 2. Immunocompromised patients and treatments.**

| | Category | 0 | | 1 | 2 |
|---|---|---|---|---|---|
| | Subjects (n) | No active treatment | Low | Medium | High |
| Hematologic cancer patients | 13 | 2 | 2 | 1 | 8 |
| SARS-CoV-2 naturally infected | | | | | |
| No | 8 | 0 | 0 | 1 | 7 |
| Yes | 5 | 2 | 2 | 0 | 1 |
| Tumor type | | | | | |
| Diffuse large B-cell lymphoma (DLBCL) | 4 | 1 | 0 | 0 | 3 |
| Hodgkin lymphoma (HL) | 1 | 0 | 0 | 1 | 0 |
| Follicular lymphoma (FL) | 5 | 0 | 0 | 0 | 5 |
| Multiple myeloma (MM) | 1 | 0 | 1 | 0 | 0 |
| Chronic lymphocytic leukemia (CLL) | 1 | 1 | 0 | 0 | 0 |
| Chronic myeloid leukemia (CML) | 1 | 0 | 1 | 0 | 0 |
| Solid cancer patients | 17 | 6 | 8 | 3 | 0 |
| SARS-CoV-2 naturally infected | | | | | |
| No | 4 | 2 | 0 | 2 | 0 |
| Yes | 13 | 4 | 8 | 1 | 0 |
| Tumor type | | | | | |
| Breast cancer | 7 | 2 | 5 | 0 | 0 |
| Lung cancer | 2 | 0 | 2 | 0 | 0 |
| Sarcoma | 6 | 4 | 0 | 2 | 0 |
| Pancreatic cancer | 1 | 0 | 1 | 0 | 0 |
| Testicular cancer | 1 | 1 | 0 | 0 | 0 |
| Rheumatic disease patients | 48 | 5 | 4 | 26 | 13 |
| SARS-CoV-2 naturally infected | | | | | |
| No | 47 | 5 | 3 | 26 | 13 |
| Yes | 1 | 0 | 1 | 0 | 0 |
| Diagnosis | | | | | |
| Autoimmune hepatitis (AIH) | 2 | 2 | 0 | 0 | 0 |
| Psoriatic arthritis/spondyloarthritis/ankylosing spondylitis (PA/SpA/AS) | 18 | 0 | 0 | 17 | 1 |
| Rheumatoid arthritis (RA) | 12 | 1 | 0 | 8 | 3 |
| Primary biliary cholangitis (PBC) | 5 | 1 | 4 | 0 | 0 |
| Sclerosing cholangitis (SC) | 1 | 0 | 0 | 1 | 0 |
| Dermatomyositis (DM) | 2 | 0 | 0 | 0 | 2 |
| Systemic lupus erythematosus (SLE) | 1 | 0 | 0 | 0 | 1 |
| Primary Sjögren's syndrome (pSS) | 1 | 0 | 0 | 0 | 1 |
| Systemic sclerosis (SSc) | 6 | 1 | 0 | 0 | 5 |
| Dialysis patients | 53 | 0 | 15 | 16 | 22 |
| SARS-CoV-2 naturally infected | | | | | |
| No | 48 | 0 | 12 | 16 | 20 |
| Yes | 5 | 0 | 3 | 0 | 2 |
| Acute kidney injury (AKI) causes | | | | | |
| ANCA-associated vasculitis | 1 | 0 | 0 | 0 | 1 |

**Table 2. Continued**

| | Category | 0 | | 1 | 2 |
|---|---|---|---|---|---|
| | Subjects (n) | No active treatment | Low | Medium | High |
| Chronic Glomerulonephritis (CGN) | 6 | 0 | 1 | 2 | 3 |
| Glomerulopathy after liver transplantation | 1 | 0 | 0 | 0 | 1 |
| Nephrolithiasis | 1 | 0 | 1 | 0 | 0 |
| Nephropathy | 5 | 0 | 0 | 3 | 2 |
| Nephrosclerosis | 33 | 0 | 9 | 11 | 13 |
| Non-Hodgkin lymphoma (NHL) | 1 | 0 | 0 | 0 | 1 |
| Polycystic kidney disease (PKD) | 5 | 0 | 4 | 0 | 1 |

Clinical information and treatments of patients with cancer (hematologic or solid cancer), rheumatic disease, or undergoing hemodialysis. Classification in categories (0, 1, and 2) is reported. Cancer and rheumatic disease patients were classified according to the type of treatment at the time of vaccination: no active treatment or low (0), medium (1), or high (2) interference with the immune system. Patients in hemodialysis were classified with an immunoscore related to the disease for which the patients are in dialysis and their comorbidities: low (0), medium (1), or high (2) immune compromised.

observed only in three of seven patients under anti-CD20 treatment at T2. Interestingly, the peripheral blood T-cell response dropped 3 mo after vaccination in a good proportion of subjects, including HCW, and in 9 of 23 (Ag1) and in 5 of 23 (Ag2) was below the threshold of positivity selected for this study. The booster dose to cancer patients re-elevated the T-cell response to levels similar to those after the second dose but we did not observe further enhancement like that of the antibody response. Anti-CD20 treated patients that did not show a T-cell response after the second dose, did not benefit from the booster dose (Fig S2A). As shown in Fig S2B, the T-cell response was boosted in all of naturally infected subjects at T2, regardless of being HCW or cancer patients with or without treatment (even anti-CD20), and it was high at 3–4 mo after vaccination (T3) or at 2 wk after the booster dose (T4).

### The immune response is compromised in a substantial proportion of patients in hemodialysis and in some rheumatologic patients but can be boosted by a third vaccine dose

Prompted by the intriguing results on cancer patients and the dependence of the immune response on the pharmacologic treatment, we evaluated whether other categories of immunocompromised patients displayed a compromised immune response to the vaccine and the outcome after a booster dose. Thus, we tested the trimeric antibody levels, their neutralization ability and T-cell responses at 2–3 mo from the second dose (T3) and at 2 wk after the booster dose (T4) in patients with rheumatic diseases or in patients in hemodialysis. As patients in hemodialysis were older, we also included a group of elderly people (≥75 y) receiving the vaccine. As shown in Fig 2A, patients in hemodialysis had a significant reduction in trimeric antibody response at 3 mo after the second dose of vaccine (T3) compared with health-care workers (P < 0.0001) and a drastic but not significative reduction versus older subjects. This response reflected also a significant reduction (P < 0.0001) in the neutralizing ability of the antibodies (Fig 2B) also in older subjects (P = 0.0026). Rheumatic disease patients instead, as a group, had a reduction in IgG trimeric antibody response, which was not statistically significant; however, the neutralization potential was significantly reduced (P = 0.0499) as compared with that of HCW

individuals (Fig 2A and B). Notably, four patients had no neutralizing antibodies, although two of them had a positive antibody test. As shown in Fig 2C and D, the T-cell response (both Ag1 and Ag2) was significantly lower as compared with HCW in hemodialysis patients (Ag1, P = 0.0003; Ag2, P = 0.0017), but not in the other patients. When we analyzed the response at 2 wk after the third dose (T4), we observed that all rheumatic patients and dialysis patients (except for one patient of each class) had increased the serum levels of antibodies (P < 0.0001) which were also neutralizing except for three patients in hemodialysis, two of them having detectable trimeric antibodies (54.2 and 134 binding antibody unit [BAU]/ml) which were not neutralizing (Fig 2A and B). However, although the T-cell response was boosted, with a statistically significant increase only in dialysis patients (Ag1, P = 0.0014; Ag2, P = 0.0015), it remained below the limit of positivity set in this study for many patients (Fig 2C and D). Importantly, in Fig S3A we reported INF-γ basal levels that may be produced by other cell types (e.g., NK cells), and that we found to be below the cut-off threshold for most samples. As observed also for cancer patients, previously exposed to SARS-CoV-2 patients displayed the highest levels of neutralizing antibodies which remained high also after the booster dose (Fig S4A and B). Moreover, the T-cell responses remained higher in SARS-CoV-2–experienced patients than naïve HCW (Figs S4C and D and S3A).

### The immune response depends on the type of treatment or immune status of the patients

Having observed a clear reduction in antibody levels in cancer or hemodialysis patients and in some rheumatologic disease patients, we analyzed whether the observed differences were linked to an immune depressed state induced by the treatment or by their disease. As described in the methods section, we classified the patients according to the type of treatment (cancer and rheumatic disease patients) or an immunoscore related to the disease for which the patients are in dialysis and their comorbidities. As shown in Figs 3A, S5A and B, and S6A and B the type of treatment (no treatment or low [0], medium [1], or high [2] interference with the immune system) or the worsening of the immunoscore in hemodialysis patients (low [0], medium [1], or high [2] immune

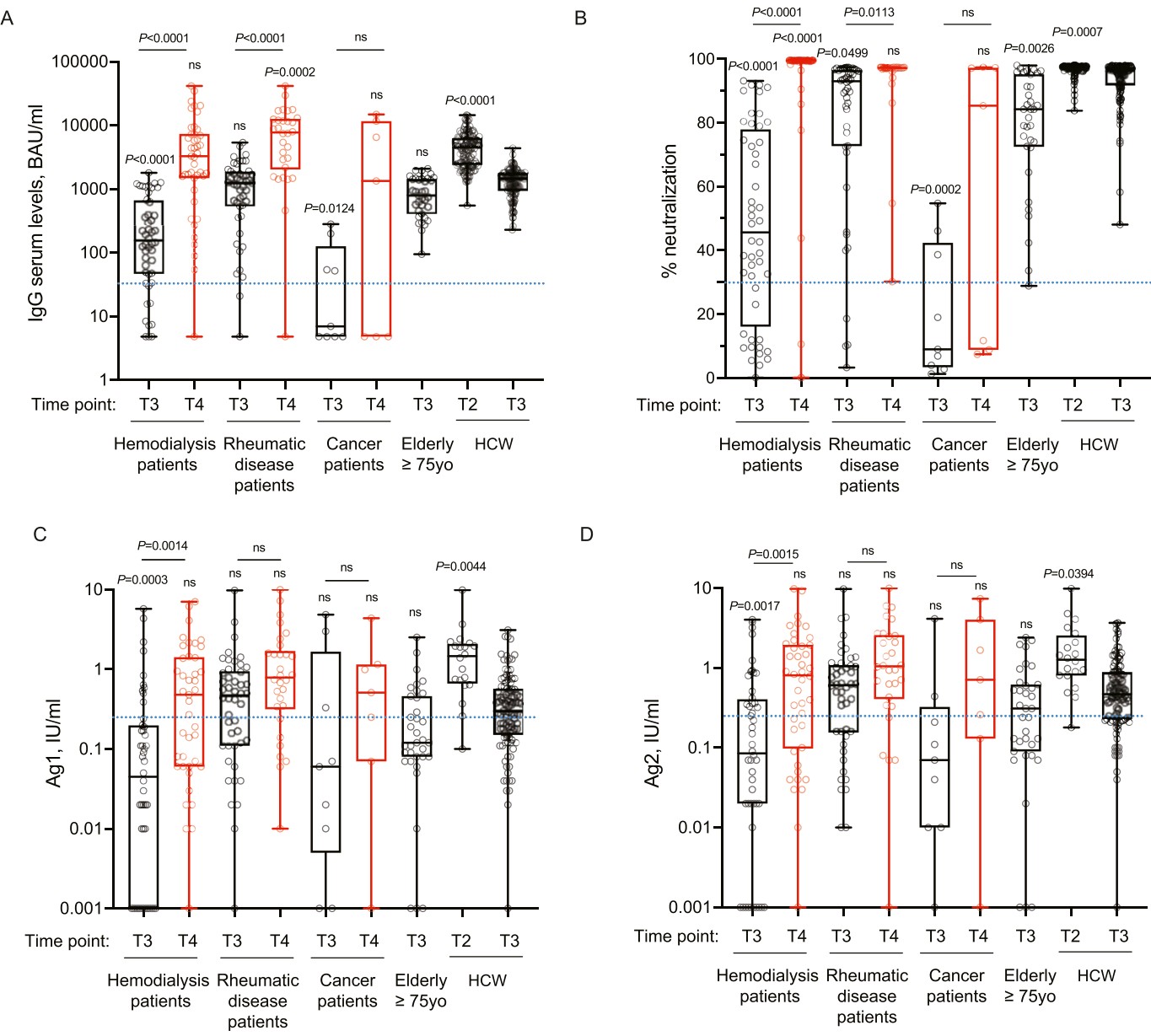

**Figure 2. The immune response is compromised in a substantial proportion of naïve patients in hemodialysis and in some naïve rheumatologic patients but can be boosted by a third vaccine dose.**
**(A, B, C, D)** IgG antibody response (A), its neutralizing activity (B) and anti-spike T-cell response activation, by using specific CD4 (Ag1, C) and CD4 plus CD8 (Ag2, D) T-cell epitopes of the spike protein were measured in serum and plasma of vaccinated naïve health-care workers (HCW, n = 104), elderly people ≥75 yr (n = 35), cancer patients (n = 9), patients with rheumatic diseases (n = 47) or patients in hemodialysis (n = 48) at 2–4 mo after second dose (black, T3), and in serum and plasma of cancer patients (n = 7), patients with rheumatic diseases (n = 30), or patients in hemodialysis (n = 43) 2 wk after the booster dose (red, T4). As a control, we indicated values of IgGs, their neutralizing activity and anti-spike T-cell response activation of vaccinated naïve health-care workers (HCW, n = 119) at 10 d after the second dose (T2). The box plots show the interquartile range, the horizontal lines show the median values, and the whiskers indicate the minimum-to-maximum range. Each dot corresponds to an individual subject. *P*-values were determined using two-tailed Kruskal–Wallis test with Dunn's multiple comparisons post-test. *P*-values refer to HCW T3 when there are no connecting lines. Positivity was based on anti-spike IgG ≥ 33.8 BAU/ml (LIAISON SARS-CoV-2 TrimericS IgG), neutralization (Neu) ≥ 30% (cPass SARS-CoV-2 Neutralization Antibody Detection Kit), and T-cell response ≥ 0.25 IU/ml for either Ag1 or Ag2 (QuantiFERON SARS-CoV-2 assay).
Source data are available for this figure.

compromised) impacted on the profile of the immune response with a progressive reduction of both antibody levels (Figs 3A, S5A and B, and S6A and B) and neutralization potential (Fig 3A). Interestingly, patients distributed quite homogenously in the three categories suggesting that their immune status, rather than the disease itself, was responsible for the impaired immune response. Particularly affected were patients belonging to category 2: patients in hemodialysis with an high immune compromised immune score, rheumatic disease patients treated with mycophenolate or methotrexate and cancer patients treated with anti-CD20 (Fig 3A,

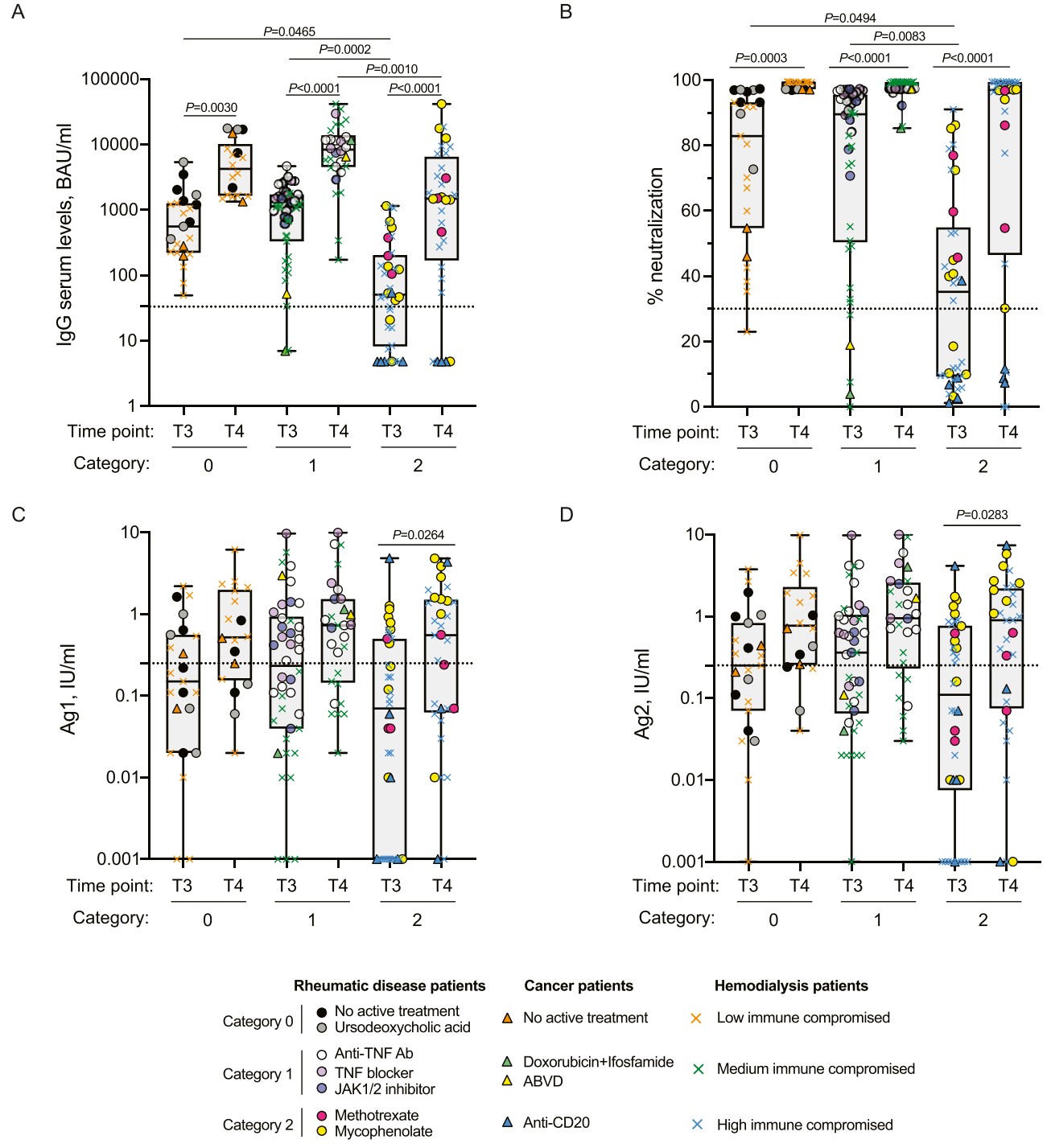

**Figure 3. The immune response depends on the type of treatment or immune status of the patients.**
**(A, B, C, D)** IgG antibody response (A), its neutralizing activity (B) and anti-spike T-cell response activation, by using specific CD4 (Ag1, C) and CD4 plus CD8 (Ag2, D) T-cell epitopes of the spike protein were measured in serum and plasma of vaccinated naïve patients with cancer (n = 9), rheumatic diseases (n = 47), or patients in hemodialysis (n = 48) at 2–4 mo after second dose (T3) and 2 wk after the booster dose (T4). Cancer and rheumatic disease patients were classified according to the type of treatment: no active treatment or low (category 0), medium (category 1), or high (category 2) interference with the immune system, whereas patients in hemodialysis were classified with an immunoscore related to the disease for which the patients are in dialysis and their comorbidities: low (category 0), medium (category 1), or high (category 2) immune compromised. The distribution of patients in each category and the type of treatment are indicated in the legend. Samples ≥33.8 BAU/ml (IgG plasma levels) or ≥30% signal inhibition (neutralization) and T-cell response ≥0.25 IU/ml for either Ag1 or Ag2 were considered positive (dotted black lines). The box plots show the interquartile range, the horizontal lines show the median values, and the whiskers indicate the minimum-to-maximum range. Each dot corresponds to an individual subject. *P*-values were determined using two-tailed Kruskal–Wallis test with Dunn's multiple comparisons post test. *P*-values are reported. Source data are available for this figure.

blue crosses, yellow or pink circles and blue triangles, respectively; Figs S5A and B and S6A and B). It should be noted that patients under methotrexate stopped treatment 1 wk after getting vaccinated and indeed they all developed neutralizing antibodies (Fig 3A and B, pink circles). Interestingly, the third dose (T4) allowed patients in category 2 to achieve levels of antibodies similar to those in category 1 at 2/4 mo after the second dose (T3) except for patients treated with anti-CD20 antibody (Fig 3A, blue triangles), one patient in hemodialysis (Fig 3A, blue cross), and one patient with mycophenolate (Fig 3A, yellow circle). Interestingly, the latter patient was advised to stop treatment for 1 wk after vaccination, but did not follow the advice. Patients in hemodialysis that were previously exposed to SARS-CoV-2 showed higher levels of IgG compared with those that were not infected, even after 6 mo from the second dose, at the time of booster (Fig S6C and D). The booster dose increased significantly the neutralization ability of IgG in all the categories (Fig 3B). The T-cell response also was affected particularly by the category of drugs with high interference with the immune system or by an immune compromised status (category 2), but differences with patients belonging to category 0 or 1 were not striking at T3. Interestingly, T-cell response (both to Ag1 and Ag2) was statistically significantly boosted with a third dose only in category 2 patients (Fig 3C and D). Moreover, in Fig S3B, we reported INF-γ basal levels, which we found to be below the cut-off threshold for most samples. Interestingly, when analyzing the correlation between antibody levels and neutralization potential, we found that in the group of patients in the category 2 (treated with drugs with high interference with the immune system or immune compromised patients) levels of trimeric antibody above 100 BAU/ml after the second dose are most likely to correspond to a positive neutralization test (>30%) (Fig 4A). The booster dose allowed most of the patients achieve a neutralizing antibody response, and it was confirmed that a level of antibodies above 100 BAU/ml correlated with a positive neutralization test (Fig 4B).

# Discussion

Here we show that, upon vaccination, elderly subjects and patients under treatments that have little or no interference with the immune system develop an immune response which is slightly reduced but comparable to that of healthy individuals, whereas those immunosuppressed (with an immunoscore equal to 2) or under immunosuppressive treatments are strongly impaired in the ability to activate an antibody response (i.e., cancer patients treated with anti-CD20 therapy or rheumatic disease patients under active treatment of mycophenolate). In some cases, the immune response is not initiated at all. However, a third booster dose allows to achieve levels of neutralizing antibodies similar to those of HCW after the second vaccine dose (T2) except for anti-CD20 treated cancer patients. By stratifying patients according to treatment, we show that anti-CD20 and mycophenolate are the drugs with the highest impact on the development of a correct immune response. By contrast, methotrexate which is associated with specific immune inhibitory drugs did not have a major impact on the immune response, but it has to be considered that methotrexate therapy was

stopped for 1 wk after every dose of vaccine, whereas mycophenolate was not interrupted at the time of the first and second dose vaccination. This suggests that, wherever possible, treatment having an impact on the immune system should be interrupted or delayed to favor the development of an immune response. Indeed, at the administration of the third dose, mycophenolate was interrupted and this resulted in a proficient activation of the immune response. The patient who did not follow the advice of interrupting mycophenolate resulted in an undetectable antibody and T-cell response even after the third booster dose, confirming that treatment should be stopped to favor the development of an immune response. Interestingly, as expected the antibody response to the trimeric form of Spike was undetectable in individuals under anti-CD20 treatment, and the situation did not change after the third dose. Interestingly also a patient that had interrupted anti-CD20 5 mo earlier still did not display antibodies to the Spike trimeric protein. This is in line with a recent report showing that patients with B-cell lymphoma receiving B-cell–directed therapies should be vaccinated at least 9 mo from the last treatment to improve antibody titers (Ghione et al, 2021). By contrast the T-cell response to AG1 and AG2 spike peptides was observed in three of seven patients under anti-CD20 treatment at T2. This to us was unexpected as it has been shown that anti-CD20–treated multiple sclerosis patients had a similar ability to induce T cells to the spike protein as healthy subjects (Apostolidis et al, 2021). This suggests that cancer patients may have an additional impairment in inducing the T-cell response which is probably unrelated to the active treatment. This makes cancer patients a very vulnerable category that needs further attention. It would be important to correlate the vaccine immune response to the stage of disease as the immune system may be depressed as a consequence of the immunosuppressive status generated by the cancer itself. Indeed, it has been shown that COVID-19 mortality was statistically significantly higher in cancer patients with an active disease (Pinato et al, 2020). Also, the immune status of the patients is strongly correlated with the ensued immune response as indicated by the impact of disease and immunoscore of patients in hemodialysis.

Regarding T-cell analysis, we decided to test the T-cell response by restimulating whole blood cells with specific peptides because we analyzed a population comprising immunocompromised individuals. Indeed, it is technically challenging to isolate T cells from immunocompromised individuals unless a large amount of blood is collected. It was already very difficult to recruit immunocompromised patients because of their disease and treatments. In addition, many of them are continuously subjected to blood draws or treatments that require intravenous access and it was unlikely that they may participate in a protocol asking to donate more blood. This was a limitation along with the inability to measure the differences in T-cell frequencies among participants. Moreover, we cannot exclude that the IFN-γ measured after stimulation with SARS-COV-2–specific peptides could be produced also by other cell types (e.g., NK cells) and that the differences observed between groups could be accounted for by differences in T-cell skewing related to the disease state and/or infection history. However, the finding that basal levels of IFN-γ were below the threshold suggests that the observed production was due to peptide restimulation.

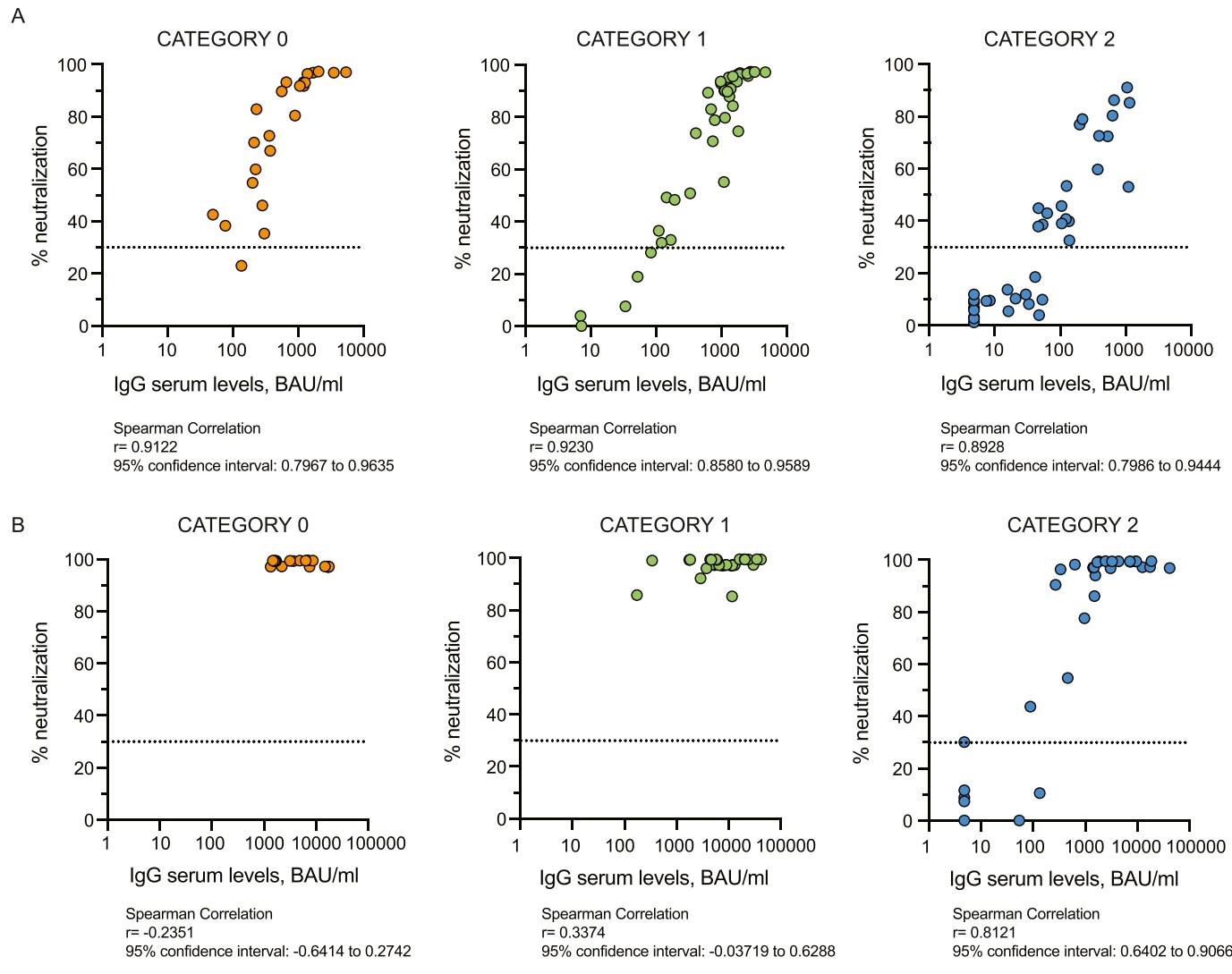

**Figure 4. Correlation between antibody levels and neutralization potential.**
**(A, B)** Correlation between IgG values in serum (x variable) and the % of neutralization (y variable) was performed in each category of immunocompromised patients at T3 (0, n = 23; 1, n = 43; 2, n = 38) (A) and at T4 (0, n = 18; 1, n = 30; 2, n = 32) (B). A nonparametric Spearman's rank correlation test was performed. Samples ≥33.8 BAU/ml (IgG plasma levels) or ≥30% signal inhibition (neutralization, dotted black line) were considered positive. Log scale on x-axis.
Source data are available for this figure.

In conclusion, immunocompromised patients should be tested periodically to assess the development and status of an immune response and should be considered individually and on the basis of their active treatments with regards to a potential booster dose. Those that are not immunized should be prioritized to receive a booster dose of vaccine and be re-evaluated afterwards for effective immunization. However, the therapeutic schedule should be modulated (interrupted or delayed) to favor an immune response to the vaccine. Particular attention should be given to patients with antibody levels below 100 BAU/ml because these antibodies are unlikely to exert a neutralizing activity. A different scenario is observed in patients previously exposed to SARS-CoV-2. These patients reach maximal response after two doses of vaccine, and still one subject under anti-CD20 treatment failed to activate an antibody response but developed a T-cell response. More

SARS-CoV-2–exposed patients should be tested with immunosuppressive treatments to draw conclusions.

# Materials and Methods

### Study design

We tested the IgG antibody response, the CD4 and CD8 T-cell activation and the neutralizing antibody response to SARS-CoV-2 spike protein developed after mRNA SARS-CoV-2 vaccination (Spikevax or Moderna mRNA-1273 – Comirnaty or BNT162b2 Pfizer-BioNTech) as a part of two observational studies approved by the Ethical Committee of Istituto Clinico Humanitas, in compliance with

the Declaration of Helsinki principles. The studies were conducted at Istituto Clinico Humanitas and comprised a longitudinal sample collection, including health-care workers (n = 160) and cancer patients (n = 30) and a cross-sectional sample collection, including elderly subjects (n = 37), patients with rheumatic diseases (n = 48), and patients in hemodialysis (n = 53). Immunocompromised patients received also a third dose (booster) ±5 mo after the second dose.

Analyzed time points were: as follows the day of the first dose (T0), 21–28 d after the first dose (T1), 10–26 d after the second dose (T2), 2–4 mo after the second dose (T3), and 2 wk after the third dose (T4).

At each scheduled time point, as shown in Fig 1, serum and lithium-heparin whole blood samples were collected from enrolled individuals. Study inclusion criteria included a vaccination with an authorized COVID-19 vaccine (according to Italian regulation and guidelines), age of 18 yr or greater, and willingness and ability to provide informed consent. Study exclusion criteria included lack of willingness and ability to provide informed consent, or a lack of properly collected and stored samples. Demographic and clinical information for healthy subjects (health-care workers and elderly) and patients can be found in Tables 1 and 2. Experiments were conducted in a blinded fashion with designated members of the clinical team, who did not run the assays, having access to the sample key until data were collected, at which point researchers of the team were unblinded. All individuals enrolled in the studies provided an informed consent as part of the protocols (CLI-PR-2102 and CLI-PR-2108). These studies began in February 2021 (CLI-PR-2102) and June 2021 (CLI-PR-2108) and are continuing with participant's follow-up. Enrolled individuals did not receive compensation for their participation.

### Patients and treatments

Cancer and rheumatic disease patients were classified according to the type of treatment: no active treatment or low (category 0), medium (category 1), or high (category 2) interference with the immune system (Table 2). In particular, drugs with low interference with the immune system (category 0) included: Tyrosine Kinase Inhibitor, TKI (Imatinib), EGFR TKI (Osimertinib), chemotherapy (Lenalidomide, Docetaxel, Gemcitabine, Nab-paclitaxel), hormone therapy, anti-HER2 agents (Pertuzumab, Trastuzumab), chemotherapy + anti-PDL1 (Carboplatin+etoposide+Atezolizumab) (for cancer patients), and ursodeoxycholic acid for rheumatic diseases patients; drugs with medium interference with the immune system (category 1) were: Doxorubicin (with Cisplatin or with Ifosfamide or present in ABVD) (for cancer patients), and anti-TNF Ab—Infliximab, Certolizumab, Adalimumab, Golimumab; TNF blocker—Etanercept; JAK1/2 inhibitor—Baricitinib; CD80/CD86 blocker–Abatacept (for rheumatic diseases patients); immunosuppressive drugs (category 2) were rituximab, Obinutuzumab (for cancer patients), mycophenolate and methotrexate in combination with immune inhibitory drugs (for rheumatic disease patients). Treatment with methotrexate or Baricitinib (JAK1/2 inhibitor) was stopped 1 wk after every dose of vaccine, whereas treatment with mycophenolate was stopped 1 wk only after the third dose of vaccine.

Patients in hemodialysis were classified with an immunoscore related to the disease for which the patients are in dialysis and their comorbidities: low (category 0), medium (category 1), or high (category 2) immune compromised (Table 2).

### Detection of SARS-CoV-2–specific IgG antibodies

Serum samples were tested using LIAISON SARS-CoV-2 TrimericS IgG (DiaSorin), a quantitative CE-marked assay for the detection of IgG antibodies recognizing the native trimeric Spike glycoprotein of SARS-CoV-2 (Bonelli et al, 2021). According to the manufacturer's instruction for use, the presence of an immune response in vaccine recipients was 100.0% (95% CI 96.3–100.0%) in 102 samples collected after ≥21 d from second dose. The levels of IgG antibodies were originally expressed in AU/ml. Following the definition of the WHO International Standard for anti-SARS-CoV-2 Immunoglobulin (NIBSC 20:136), the readout was updated and the assay currently calculates the levels of SARS-CoV-2 IgG antibodies in BAU/ml (Perkmann et al, 2021). Samples ≥33.8 BAU/ml were considered positive. In Fig S6, for the determination of IgG anti–SARS-CoV-2 in the serum of patients in hemodialysis the Liaison SARS-CoV-2 S1/S2 IgG assay (DiaSorin) was used (Bonelli et al, 2020).

### SARS-CoV-2 neutralization assay

Neutralization was assessed by ELISA with cPass SARS-CoV-2 Neutralization Antibody Detection Kit (GenScript), a qualitative CE-marked assay for the detection of circulating neutralizing antibodies that block the interaction between the receptor binding domain of the viral spike glycoprotein with the ACE2 cell surface receptor (Tan et al, 2020). Samples were analyzed following the manufacturer's instruction for use. Samples ≥30% signal inhibition were considered positive.

### Detection of SARS-CoV-2–specific cell-mediated immunity

T-cell–mediated responses were analyzed using QuantiFERON SARS-CoV-2 Research Use Only assay (QIAGEN), following the manufacturer's instruction for use. We tested the IFN-γ production before and after restimulation with SARS-CoV-2–specific antigens. Briefly, fresh whole blood samples were collected in lithium-heparin tubes and maintained at room temperature for no more than 16 h from the time of collection. Each blood sample was transferred in a NIL-Tube (without antigens: this sample indicate the IFN-γ basal level, before restimulation) and in two QuantiFERON SARS-CoV-2 blood collection tubes containing different cocktails of SARS-CoV-2–specific antigens (Ag1 and Ag2) and incubated at 37°C for 16–24 h. Plasma samples retrieved after centrifugation at 2,700g at room temperature for 15 min were analyzed using LIAISON XL instrument (DiaSorin) for detection of IFN-γ, according to the standard procedures recommended by the manufacturer. For this study, positive results were defined as ≥0.25 IU/ml, after IFN-γ basal level (NIL tube) was subtracted from Ag1 and Ag2 values. In Fig S3 we showed the IFN-γ basal level (IU/ml). We defined this tentative cut-off threshold based on previous experience with the QuantiFERON test but this is arbitrary as other studies have defined a lower cut off between 0.15 and 0.2 (Van Praet et al, 2021).

## Statistical analysis

Data were analyzed for normal distribution (Shapiro–Wilk test) before any statistical analyses. Individual values are presented as spaghetti plots or as box plots showing the interquartile range, median, and minimum-to-maximum whiskers. The differences between matched time points were analyzed using the nonparametric Friedman test with Dunn's multiple comparisons test. The comparison of multiple groups was carried out using the nonparametric Kruskal–Wallis test followed by Dunn's multiple comparisons test. To gauge the correlation between IgG values in plasma (x variable) and the % of neutralization (y variable), a nonparametric Spearman's rank correlation test was performed. A probability value of $P < 0.05$ was considered significant. All statistics and reproducibility information are reported in the figure legends. Data analyses were carried out using GraphPad Prism version 8.

## Supplementary Information

## Acknowledgements

The reagents in this study were supplied by DiaSorin (Italy) and Quanti-FERON SARS-CoV-2 tubes by QIAGEN (Germany). We acknowledge the full financial support of DiaSorin S.p.A. for CLI-PR-2102 and CLI-PR-2108 clinical trials. In DiaSorin we would like to thank Elisa Ghezzi, Clara Rossini, and Chiara Mauro for neutralization testing and Alice Bianchi for T-cell response testing. We thank Jenny Howard, Francis Stieber, and Vladyslav Nikolayevskyy from QIAGEN for their scientific support during the study. We would like to thank all the employees and the patients that volunteered to participate to this study, all the vaccinating doctors, the nurses and personnel that collected the samples, the laboratory technicians that run the serological tests, and Humanitas Operations Management and Customer Care that coordinated vaccinations and blood draws. This work was supported by DiaSorin S.p.A. and Ricerca Corrente (Ministero della Salute, to M Rescigno and A Mantovani).

### Author Contributions

E Azzolini: conceptualization and project administration.
C Pozzi: data curation, formal analysis, project administration, and writing—original draft, review, and editing.
L Germagnoli: data curation, formal analysis, and methodology.
B Oresta: data curation and project administration.
N Carriglio: data curation and project administration.
M Calleri: conceptualization, resources, and project administration.
C Selmi: supervision and writing—review and editing.
M De Santis: data curation, investigation, and writing—review and editing.
S Finazzi: data curation, investigation, and writing—review and editing.
C Carlo-Stella: data curation, investigation, and writing—review and editing.
A Bertuzzi: data curation, investigation, and writing—review and editing.
F Motta: data curation, investigation, and writing—review and editing.
A Ceribelli: data curation, investigation, and writing—review and editing.
A Mantovani: conceptualization, supervision, funding acquisition, and writing—review and editing.
F Bonelli: conceptualization, resources, supervision, funding acquisition, and project administration.
M Rescigno: conceptualization, supervision, funding acquisition, and writing—original draft, review, and editing.

### Conflict of Interest Statement

F Bonelli, M Calleri, N Carriglio, and B Oresta are employees of DiaSorin S.p.A., the manufacturer of the LIAISON SARS-CoV-2 TrimericS IgG test. Employees of DiaSorin participated in the study design, data collection, and interpretation and in the preparation of the manuscript. M Rescigno participated to advisory boards and received support from Diasorin S.p.A. Other authors have declared that no conflict of interest exists.

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
