## [Reviewer comments · Life Science Alliance]

Life Science Alliance

mRNA COVID-19 vaccine booster fosters B and T cell responses in immunocompromised patients.

Elena Azzolini, Chiara Pozzi, Luca Germagnoli, Bianca Oresta, Nicola Carriglio, Mariella Calleri, Carlo Selmi, Maria De Santis, Silvia Finazzi, Carmelo Carlo-Stella, Alexia Bertuzzi, Francesca Motta, Angela Ceribelli, Alberto Mantovani, Fabrizio Bonelli, and Maria Rescigno

DOI: <https://doi.org/10.26508/lsa.202201381>

Corresponding author(s): *Maria Rescigno, Humanitas University*

Review Timeline:

Submission Date:	2022-01-21
Editorial Decision:	2022-01-21
Revision Received:	2022-02-02
Editorial Decision:	2022-02-03
Revision Received:	2022-02-04
Accepted:	2022-02-07

Transaction Report:

Please note that the manuscript was previously reviewed at another journal and the reports were taken into account in the decision-making process at *Life Science Alliance*. Since the original reviews are not subject to Life Science Alliance's transparent review process policy, the reports and author response cannot be published.

January 21, 2022

Re: Life Science Alliance manuscript #LSA-2022-01381-T

Prof. Maria Rescigno
Humanitas University
Department of Biomedical Sciences
Pieve Emanuele, MI
Italy

Dear Dr. Rescigno,

Thank you for submitting your manuscript entitled "A booster dose of mRNA-based COVID-19 vaccines fosters the development of an immune response in immunosuppressed fragile patients" to Life Science Alliance. We invite you to re-submit the manuscript, revised to address the Reviewer comments as you've outlined.

Please note that papers are generally considered through only one revision cycle.

Thank you for this interesting contribution to Life Science Alliance. We are looking forward to receiving your revised manuscript.

Sincerely,

B. MANUSCRIPT ORGANIZATION AND FORMATTING:

February 3, 2022

RE: Life Science Alliance Manuscript #LSA-2022-01381-TR

Prof. Maria Rescigno
Humanitas University
Department of Biomedical Sciences
Via Rita Levi Montalcini, 4
Pieve Emanuele, MI 20072
Italy

Dear Dr. Rescigno,

Thank you for submitting your revised manuscript entitled "mRNA COVID-19 vaccine booster fosters B and T cell responses in immunocompromised patients.". We would be happy to publish your paper in Life Science Alliance pending final revisions necessary to meet our formatting guidelines.

- please upload your main manuscript text as an editable doc file;
- please upload your main and supplementary figures as single files;
- please add ORCID ID for the corresponding author-you should have received instructions on how to do so
- please add the Twitter handle of your host institute/organization as well as your own or/and one of the authors in our system
- please add your main, supplementary figure, and table legends to the main manuscript text after the references section
- we encourage you to revise the figure legend for figure S3 such that the figure panels are introduced in alphabetical order
- please upload your Tables in editable .doc or excel format
- please add callouts for Figures S5A-B and S6A-B to your main manuscript text

A. FINAL FILES:

B. MANUSCRIPT ORGANIZATION AND FORMATTING:

Sincerely,

February 7, 2022

RE: Life Science Alliance Manuscript #LSA-2022-01381-TRR

Prof. Maria Rescigno
Humanitas University
Department of Biomedical Sciences
Via Rita Levi Montalcini, 4
Pieve Emanuele, MI 20072
Italy

Dear Dr. Rescigno,

Thank you for submitting your Research Article entitled "mRNA COVID-19 vaccine booster fosters B and T cell responses in immunocompromised patients.". It is a pleasure to let you know that your manuscript is now accepted for publication in Life Science Alliance. Congratulations on this interesting work.

DISTRIBUTION OF MATERIALS:

Again, congratulations on a very nice paper. I hope you found the review process to be constructive and are pleased with how the manuscript was handled editorially. We look forward to future exciting submissions from your lab.

Sincerely,
